# FORMAL SPECIFICATIONS FROM NATURAL LANGUAGE

## ABSTRACT

We study the generalization abilities of language models when translating natural language into formal specifications with complex semantics. In particular, we fine-tune language models on three datasets consisting of English sentences and their corresponding formal representation: 1) regular expressions (regex), frequently used in programming and search; 2) First-order logic (FOL), commonly used in software verification and theorem proving; and 3) linear-time temporal logic (LTL), which forms the basis for industrial hardware specification languages. Our experiments show that, in these diverse domains, the language models maintain their generalization capabilities from pre-trained knowledge of natural language to generalize, e.g., to new variable names or operator descriptions. Additionally, they achieve competitive performance, and even outperform the state-of-the-art for translating into regular expressions, with the benefits of being easy to access, efficient to fine-tune, and without a particular need for domain-specific reasoning.

## 1 INTRODUCTION

Translating natural language into *formal* languages is a long-standing goal of artificial intelligence research dating back to the 1960s (e.g., Weizenbaum (1966); Winograd (1971)). Due to recent progress in deep learning (especially Vaswani et al. (2017)) and the development of language models (LMs), the field has seen significant improvements, for instance, in the translation from natural language into coding languages or formal mathematics (e.g., Lewkowycz et al. (2022); Chowdhery et al. (2022); Chen et al. (2021); Wu et al. (2022)). In this paper, we study the generalization abilities of a pre-trained LM when translating natural language into *formal specification languages*.

Formal specification languages are used in various computer science fields to describe a system's desired behavior, including fields such as systems design, requirements analysis, and automated reasoning. Examples include specification languages based on logics, such as Alloy (Jackson, 2002) and LTL (Pnueli, 1977), system specification languages based on state charts, such as SDL (Fonseca i Casas et al., 2013), or text processing specifications based on regular languages, omega-regular languages, and automata theory (Aho, 1991; Thomas, 1990). Compared to natural language, the benefit of a formal specification language is its unambiguous semantics making it accessible for algorithmic work that relies on a specification as input. Examples are high-performance SAT and SMT solvers (e.g., Sorensson & Een (2005); Biere et al. (2013); Audemard & Simon (2018); Moura & Bjørner (2008); Barrett et al. (2011)), planning tools LaValle (2006), model checkers (e.g., Cimatti et al. (2002); Holzmann (1997); Behrmann et al. (2006)), hardware synthesis tools (e.g., Bohy et al. (2012); Faymonville et al. (2017); Meyer et al. (2018)), or automatic theorem provers (e.g., Bertot & Castéran (2013); Nipkow et al. (2002)). Despite their benefits and various application areas, formal specification languages are still almost exclusively used by domain experts as their application requires significant domain-specific knowledge and extensive manual work. With the success of LMs, the goal of making the techniques mentioned above available to a broader user base to increase the correctness, trust, and assurance in computer systems is finally getting closer.

So far, efforts in utilizing deep learning to translate natural language into formal specifications have relied on training (often over-engineered) neural networks from scratch (e.g., Singh et al. (2020); He et al. (2022)). Such approaches are naturally limited in their generalization capabilities. The natural questions arise: 1) Can off-the-shelf LMs achieve competitive performance when fine-tuned on this challenging translation task? 2) How well will they generalize with their pre-trained knowledge of natural language? In this work, we initiate a study on this topic by fine-tuning the open-source transformer language model T5 (Raffel et al., 2020). The transformer architecture (Vaswani et al.,

| | |
|---|---|
| natural language (ID) | lines having a character and the string 'dog' in them |
| regex prediction (correct) | ((.)&(dog).* |
| natural language (OOD) | lines with words with a letter before the string 'eye' or the string 'time' |
| regex prediction (correct) | ([A-Za-z]).*( (eye) \| (time) ).* |
| natural language (ID) | Globally it is the case that if a holds then eventually a and b hold. |
| LTL prediction (correct) | $\Box(a \rightarrow \Diamond(a \land b))$ |
| natural language (OOD) | Whenever x does not hold , o9 will eventually hold. |
| LTL prediction (correct) | $\Box (\neg\, x \rightarrow \Diamond\, o9\,)$ |

Figure 1: An ID example of a regex model trained solely on the noun "dog", tested OOD on new nouns "eye" and "time"; and an ID example of an LTL model trained on variables $i_0$ to $i_4$ and $o_0$ to $o_4$, tested OOD on new variables and operator descriptions (bottom). OOD fragments are highlighted.

2017) has proven itself to be the most powerful general-purpose model at the moment of writing, setting new standards in many application domains such as computer vision (e.g., Dosovitskiy et al. (2020)), speech recognition (e.g., Dong et al. (2018)), and, especially, natural language processing (e.g., Brown et al. (2020)). Additionally, T5 is open-source and the trained models are easily accessible to a broad audience.

We have picked three common yet diverse formal representations used widely in software and hardware domains: 1) regular expressions, frequently used in programming and text manipulation, 2) First-order logic, which is a standard formalism used in software domains, such as theorem proving, and 3) Linear-time temporal logic, which is used in hardware domains, such as model checking of sequential circuits. Regular expressions (regex), introduced by Kleene et al. (1956), are sequences commonly used for text manipulation. For example, (a|b)* reads as "all sequences with no symbols other than a and b, including the empty string". First-order logic (FOL) extends propositional logic with predicates and quantification. With the foundations developed independently by Gottlob Frege and Charles Peirce (Peirce, 1933), FOL is a formal system of high importance in mathematics, computer science, and linguistics. For example, the formula $\forall x.\exists y.\neg(x = y)$ denotes that for every $x$, there is a $y$, which is not equal to $x$. Linear-time temporal logic (LTL) (Pnueli, 1977) is a hardware specification language widely used by the verification community. It forms the basis for industrial specification languages like the IEEE standard PSL (IEEE-Commission et al., 2005). LTL extends propositional logic with temporal operators, specifying behavior over time. For example, when considering a controller for a shared resource, the formula $\Box(r \rightarrow \Diamond g)$ denotes that it is "always the case that a request $r$ is eventually followed by a grant $g$".

Our experiments show that the fine-tuned LM achieves competitive performance on all tasks and even improves state-of-the-art performance in translating natural language to regex by 6 percentage points. Additionally, the models can utilize pre-trained knowledge of natural language. For example, Figure 1 shows hand-picked in-distribution (ID) and out-of-distribution (OOD) examples for models trained on translating natural language to regex and LTL, respectively. The regex model generalizes to new nouns that were not present during fine-tuning. The LTL model was fine-tuned on "globally" and "always" as the translation of the LTL operator $\Box$, on "implies" and "if then" as the translation of the implication $\rightarrow$, and on variables $i_0$ to $i_4$ and $o_0$ to $o_4$. It generalized to new variable names and operator descriptions, recognizing $x$ and $o9$ as variables, "whenever" as a synonym for "globally", and a simple comma as a synonym for "implies". We provide detailed experiments in Section 4 showing, for example, that the regex model achieves the same accuracy on a held-out test set ($> 88\%$) when being trained on only four out of 16 occurring nouns in the test set (c.f., Figure 2 in Section 4).

In summary, we make the following contributions. We provide the first fine-tuned off-the-shelf language models for translating natural language into formal specifications, including a new state-of-the-art model for translating into regular expressions. We contribute two novel datasets for translating natural language into FOL and two for translating natural language into LTL.[1] Furthermore, we analyze the generalization capabilities of the pre-trained language models by conducting generalization experiments on new variables, nouns, and operator descriptions, as well as out-of-distribution instances.

---

[1] The datasets, models, and code will be published once the double-blind reviewing process ends.

## 2   RELATED WORK

*Natural language to regex.* Similarly to FOL, there were early rule-based techniques for regex translation (Ranta, 1998). The regex datasets have been made more amenable to translation using semantic parsing for decomposition (Kushman & Barzilay, 2013). Training has been guided towards semantically equivalent (Zhong et al., 2018) or approximately equivalent regular expressions (Park et al., 2019); the natural language descriptions have been enriched by paraphrases generated by crowdsourcing (Locascio et al., 2016). The latter work is the most closely related to ours, as it also does not use domain-specific reasoning such as, e.g., semantic equivalence. Ye et al. (2020) have proposed to solely learn generation of regex sketches, and to relegate the construction of the final, correct regular expression to a program synthesis procedure; their dataset is not publically available.

*Natural language to FOL.* The task of translating natural language into logics, for example with rule-based (e.g., Johnson (1984); Woods (1973); Thompson et al. (1969); Waltz (1978); Hendrix et al. (1978); Templeton & Burger (1983)) or statistical approaches (Zelle & Mooney, 1996; Thompson, 2003; Zettlemoyer & Collins, 2007; 2012; Kwiatkowksi et al., 2010), and recently also neural methods (Kočiský et al., 2016; Buys & Blunsom, 2017; Cheng et al., 2017; Liu et al., 2018; Li et al., 2018) has been studied extensively in the past in the area of semantic parsing Kamath & Das (2018). In this work, we rely on the FOL translation (Kamp & Reyle, 2013) of `boxer`'s output (Bos, 2015). Closest to our work on FOL translations is the first approach of translating natural language to FOL presented by Singh et al. (2020). They construct a dataset using semantic parsing, but clean up the representation of `boxer`'s FOL output, and train a highly specialized LSTM-based architecture. At the time of writing, no code or dataset are publically available for a direct comparison. Han et al. (2022) independently developed a few-shot learning approach using very large language models, achieving a similar accuracy on novel datasets.

*Natural language to LTL.* Other approaches to the problem of translating from natural language to LTL focus on the robotics domain, such as temporal aspects in grounded robotics (Wang et al., 2020) and planning (Patel et al., 2019). A survey of earlier research beyond neural approaches is provided by Brunello et al. (2019). Grammar-based approaches to translate LTL into structured natural language (Konrad & Cheng, 2005; Grunske, 2008) inspired the design of our grammar for constructing the dataset. Gavran et al. (2020) present an interactive method for translating into LTL specifications from example traces by combining SMT solving and semantic parsing. Cherukuri et al. (2022) consider the inverse direction: translating from LTL formulas to natural language.

*Deep Learning in formal reasoning tasks.* The term autoformalization (Wang et al., 2018; Szegedy, 2020; Wu et al., 2022) has been coined for tasks of translating between natural language and formal mathematics. Deep learning approaches were able to handle symbolic representations such as logical formulas in SAT-solving (Selsam et al., 2019; Selsam & Bjørner, 2019), expressions in mathematics (Lample & Charton, 2020), formalizations in theorem proving (Polu & Sutskever, 2020), specifications in hardware synthesis (Hahn et al., 2020; 2021), or even code in software generation (Li et al., 2022; Chen et al., 2021). Transformer models have succesfully been trained on programming language translation (Roziere et al., 2020), on source code to learn representations of programs (Hellendoorn et al., 2020), and on code synthesis (Li et al., 2022; Chen et al., 2021; Nijkamp et al., 2022) all lacking a training for formal representation of their specifications. Saxton et al. (2019); Schlag et al. (2019) study to solve math problems given in natural language. Transformers were also trained on symbolic integration and solving differential equations (Lample & Charton, 2020). Transformers have been applied to formal mathematics (Rabe et al., 2020).

## 3   DATA SETS

We consider three formal specification domains: 1) regular expressions (regex) frequently used in programming or search, 2) First-order logic (FOL), which is a standard formalism used in software domains, such as theorem proving, and 3) Linear-time Temporal Logic (LTL), which is used in verification, such as hardware model checking. We train on six datasets, two for each considered domain (see Table 2 in the appendix for an overview). For regular expressions, we used the existing benchmark sets `Regex-synthetic` and `Regex-turk`. The `FOL` and `LTL` datasets are new contributions. In the following, we give background on the respective domains and describe the existing datasets and our data generation methods in detail.

### 3.1 NATURAL LANGUAGE AND REGEX PAIRS

Regular expressions (regex) are sequences that describe a search pattern for natural language text. They are commonly used in programming, for example, for string-searching or find-and-replace operations. They have been introduced by Kleene et al. (1956) and are used extensively in text editors, and are even supported natively in many programming languages. For example, `(a|b)*` reads as "all sequences with no symbols other than a and b, including the empty string". We follow the regex representation defined in previous work (see Figure 5 in the appendix).

The `Regex-synthetic` dataset was synthetically generated by Locascio et al. (2016). They used a manually-crafted grammar based on the smaller dataset from Kushman & Barzilay (2013). Two randomly drawn samples from this dataset are "lines with a number or the string 'dog', zero or more times" paired with `(([0-9])|(dog))*` and "lines not starting with a character, 2 or more times" paired with `~(((.)(.*))2,)`. `Regex-turk` is a dataset that Locascio et al. (2016) generated based on paraphrases of the natural language descriptions in `Regex-synthetic`, collected through crowdsourcing at Amazon Mechanical Turk. Two randomly drawn samples from this dataset are "a letter appears before a number in the lines" paired with `.*([A-Za-z]).*([0-9]).*.*` and "lines do not start with the string 'dog' nor the string 'truck'" paired with `~(((dog)(.*))&(truck))`.

### 3.2 NATURAL LANGUAGE AND FOL FORMULA PAIRS

First-order logic (FOL) extends propositional logic with predicates and quantification. With the foundations being developed independently by Gottlob Frege and Charles Peirce (Peirce, 1933), FOL is a formal system of high importance in mathematics, computer science, and linguistics. First-order terms and formulas are defined relative to a given signature. A first-order signature is a pair of disjoint sets $\mathcal{F}$ and $\mathcal{P}$ of function and predicate symbols, respectively, as well as an arity function $\mathcal{F} \cup \mathcal{P} \rightarrow \mathbb{N}$. Given a signature, the FOL alphabet consists of the elements of $\mathcal{F}$ and $\mathcal{P}$ as well as standard logical connectives $(\neg, \vee, \wedge, \rightarrow, \top, \bot)$, quantifiers $\forall$ and $\exists$, the equality symbol $=$, and an infite set of variables $\{x_1, x_2, \ldots\}$. The syntax of a well-defined formula is given as follows:

$$t ::= x \mid c \mid f(t_1, \ldots, t_n)$$
$$\alpha ::= Q \mid P(t_1, \ldots, t_n) \mid = (t_1, t_2) \mid \top \mid \bot \mid \neg\alpha \mid \alpha_1 \wedge \alpha_2 \mid \exists x.\alpha \ ,$$

where $x$ is a variable, $c$ is a constant, $f$ is an $n$-ary function, $Q$ is a nullary predicate and $P$ an $1 \leq n$-ary predicate. The boolean connectives $\vee, \rightarrow$, and $\leftrightarrow$ as well as the quantifier $\forall$ can be derived. For example, the formula $\forall x.\exists y.\neg = (x, y)$ denotes that forall $x$, there is a $y$, which is not equal to $x$.

We generated FOL formulas from natural language sentences using the `candc` (Clark & Curran, 2004) and `boxer` (Bos, 2015) toolchain. `candc` is a wide-coverage Combinatory Categorial Grammar (CCG) parser. A CCG (Steedman, 2001) is a lexicalized grammar where every word in a sentence is assigned an elementary syntactic structure. A derivation of this CCG is then given to `boxer`, which provides a semantic framework to output various formal derivations of the input sentence, e.g., in first-order logic. Both datasets `FOL-mnli` and `FOL-codesc` are generated using this toolchain. The dataset `FOL-mnli` consists of small sentences taken from the hypothesis predictions of the glue/mnli dataset (Williams et al., 2018). Two randomly drawn examples are "The fans do not bring any support." and "No one will ever understand how continental plates form.". The dataset `FOL-codesc` consists of pairs of natural language sentences of java code snippets and their first-order translations. We sampled the pairs from the recently published Codesc (Hasan et al., 2021) dataset consisting of 4.2M datapoints. We cut off the natural language descriptions after the first sentence and translated them into an FOL formula with the `candc-boxer` toolchain. This results in a highly challenging dataset, which we believe to be close to practical applications. For example, two randomly drawn instances are "deletes a certificate from a specified key vault" and "sets the base dir for the volume".

### 3.3 NATURAL LANGUAGE AND LTL FORMULA PAIRS

Linear-time temporal logic (LTL) (Pnueli, 1977) is a temporal logic for the verification of hardware systems. LTL extends propositional logic with temporal operators, specifying behavior over time. LTL formulas are defined over a set of variables $AP$ called atomic propositions. The alphabet consists of elements of $AP$, standard logical connectives $(\neg, \vee, \wedge, \rightarrow, \top, \bot)$, and temporal operators $\bigcirc$ (next)

and $\mathcal{U}$ (until). The syntax of an LTL formula is given as follows:

$$\varphi ::= p \mid \neg \varphi \mid \varphi_1 \vee \varphi_2 \mid \bigcirc \varphi \mid \varphi_1 \, \mathcal{U} \, \varphi_2 \ ,$$

where $p \in AP$ is an atomic proposition and $\bigcirc \varphi$ means that the subformula $\varphi$ holds in the next timestep or cycle and $\varphi_1 \, \mathcal{U} \, \varphi_2$ means that $\varphi_1$ holds until $\varphi_2$ holds. We additionally use the derived operaters *eventually* $\Diamond \varphi = \top \, \mathcal{U} \, \varphi$ and *globally* $\Box \varphi = \neg \Diamond \neg \varphi$. For example, when considering a controller for a shared resource, the formula $\Box(r \to \Diamond g)$ denotes that "it is always the case that a grant to the resource $g$ eventually follows a process' request $r$".

We generated pairs of natural language sentences and LTL formulas with two different methods. In the first data generation method (`LTL-pattern`), we utilized these specification patterns commonly defined in the literature (Dwyer et al., 1998; Etessami & Holzmann, 2000; Holeček et al., 2004; Pelánek, 2007), which are provided by the `spot` library (Duret-Lutz et al., 2016). For example, the specification pattern $\Box(a \to b)$ states that at every timestep, whenever $a$ holds, $b$ has to hold as well and the specification pattern $\Box \Diamond a$ states that $a$ has to hold infinitely often. Since an LTL specification typically consists of a conjunction of such patterns, we followed the approach in the literature and conjoined up to $4$ patterns and their translations (Li et al., 2013). In the second dataset, we constructed pairs of natural language sentences and formulas using a straight-forward grammar with minimal domain-specific knowledge (Konrad & Cheng, 2005; Grunske, 2008) (see Appendix D in the appendix). The grammar restricts formulas to only contain negations directly in front of atomic propositions, which is dictated by the structure of the English language, as verbs follow a different conjugation depending on whether they are used in a positive or a negated case. For instance, $\Box a$ is translated to "Globally a holds" and $\Box \neg a$ is translated to "Globally a does not hold". To translate LTL formulas automatically, we used a natural language grammar that is structurally the same as the LTL grammar. The interested reader can find the grammar and a detailed explanation in Appendix D. The dataset `LTL-synthesis` consists of pairs of a natural language translation with our grammar (see Appendix D) and their LTL hardware synthesis specification. These hardware synthesis specifications are taken from a recently published dataset, where the authors trained a Transformer to predict hardware circuits directly from LTL specifications (Schmitt et al., 2021). The synthesis specifications consist of an LTL formula expressing the assumptions posed on the environment and an LTL formula expressing the desired guarantees of the system. They can be combined into a single LTL formula by implication.

## 4 EXPERIMENTS

We fine-tuned the `base` version of the open-source language model T5 Raffel et al. (2020) with 220 million parameters on an NVIDIA DGX A100 system for around 1 hour each run with a learning rate of $0.001$. We needed to use the `small` version for our baseline experiments on an untrained T5 model to achieve stable training. We use PyTorch (Paszke et al., 2019) and the huggingface transformers library (Wolf et al., 2020) to fine-tune the models. We report accuracy of the best-performing models (see Appendix A for ablations). In general, achieving stable training for the baseline T5 model was challenging and required much more engineering effort compared to the pre-trained version of T5 (c.f Figure 3). We split the data into $90\%$ training, $5\%$ validation, and $5\%$ test data. Table 1 summarizes the test results. We used the following prompt, respectively: `"translate natural language to {FOL | LTL | a regular expression}:"`.

### 4.1 REGULAR EXPRESSIONS

*New state-of-the-art by semantic generalization.* The fine-tuned language model achieves a new state-of-the-art in translating natural language to regular expressions on both datasets. This even holds true when comparing against state-of-the-art reinforcement learning approaches (Zhong et al., 2018; Park et al., 2019); indicated in Table 1 by (RL). A natural language sentence has multiple correct translations into a regular expression. For example, the following prediction is correct, yet different from the training target:

| | |
|---|---|
| natural language description | lines starting with a character followed by a vowel, 7 or more times |
| model prediction (correct) | ((..*[AEIOUaeiou].*){7,})(.*) |
| training target | ((..*[AEIOUaeiou].*)(.*)){7,} |

To account for such predictions, the accuracy of the regex models is evaluated with an equivalence check, called semantic accuracy Locascio et al. (2016). On the synthetically generated dataset `Regex-synthetic`, the LM achieves $94.01\%$ semantic accuracy; on the `Regex-turk` dataset, the language model achieves $64.20\%$ semantic accuracy. Due to the model's generalization to the semantic, its performance increased from $90.62\%$ to $94.01\%$ and $47.00\%$ to $64.20\%$, respectively, being the decisive factor in beating the state-of-the-art. This is exceptionally substantial on the `Regex-turk` dataset. Figure 3 (top left) depicts the accuracy per sequence of the best performing models during training. While the baseline model achieves the same accuracy (with longer training) on `Regex-synthetic`, the pre-trained model outperformes the baseline on `Regex-turk` by a significant margin. Note that we incorporate no additional training objective in contrast to previous work (Zhong et al., 2018; Park et al., 2019).

*Generalization to new nouns.* The high accuracy of the fine-tuned LM on this task poses the question if the model does "forget" its knowledge of the natural language during fine-tuning (see, e.g., He et al. (2021)). In this experiment, we tested the models generalization to English nouns that were not present during fine-tuning, but certainly during pre-training. Figure 2 shows the results of this experiment for the pre-trained T5 model (left) and the baseline T5 model (right). The first three nouns are the ones present in the datasets, i.e., "dog", "truck", and "ring". When fine-tuning on only four nouns, by adding another commonly used noun, namely "time", the model generalizes seamlessly to 16 nouns. The additional nouns were drawn from the 25 most common English nouns. Unsurprisingly, the baseline T5 model shows limited generalization capabilities to novel nouns and the pre-trained model consistently performs better, also for less nouns during training. Figure 3 (bottom right) shows the accuracies on the respective validation set. A similar observation can be made when testing on numbers that were not present during fine-tuning:

| | |
|---|---|
| natural language description | lines with the string 'dog' or a letter, 9 or more times |
| model prediction (correct) | ((dog)\|([A-Za-z])){ 9 ,} |

*OOD-testing across datasets.* As a final experiment in the regex domain, we cross-tested the models on the regex datasets. Such out-of-distribution (OOD) tests are known to be challenging for neural networks. It is especially interesting if a model trained on `Regex-synthetic`, which is purely synthetic, can translate instances of `Regex-turk`, which is constructed by humans. The model trained on the syntactic data achieved a semantic accuracy of $49.20\%$, which is only 15 percentage points behind the models accuracy that was trained on this dataset, and only 9 percentage points behind the previous state-of-the-art. Interestingly, the model can interpret ambiguous natural language sentences differently than its human counterpart and even corrects buggy targets, probably due to being trained on a slightly different dataset. For example:

| | |
|---|---|
| natural language description | lines with a number that comes before a letter, and a vowel, and the string 'dog' |
| model prediction ("incorrect") | ([0-9]).*(((([A-Za-z])&([AEIOUaeiou])&(dog)).* |
| training target | ((([AEIOUaeiou])&(dog)&([0-9])).*([A-Za-z]).* |

In the "easier" direction, the model trained on `Regex-turk` achieved an accuracy of $83.83\%$ falling only 10 percentage points short behind the model trained on this dataset and 4 percentage points

Table 1: Accuracy of the best runs for fine-tuned T5 language models on held-out test sets, where steps denote the number of training steps; accuracy is reported as the accuracy per sequence.

| dataset | previous SOTA | baseline T5 (steps) | fine-tuned T5 (steps) |
|---|---|---|---|
| Regex-synthetic | 88.7 / 91.6 (RL) | **94.01** (5K) | **94.01** (1K) |
| Regex-turk | 58.2 / 62.8 (RL) | 58.0 (5K) | **64.20** (1K) |
| FOL-mnli | **56.10** (estimated) | 46.87 (10K) | 53.91 (5K) |
| FOL-codesc | - | 58.59 (10K) | **58.98** (3K) |
| LTL-pattern | - | **100.00** (5K) | **100.00** (1K) |
| LTL-synthesis | - | 87.50 (5K) | **87.90** (1K) |

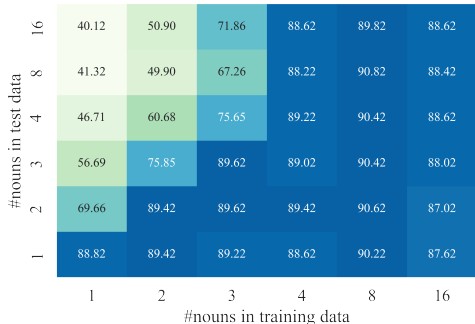 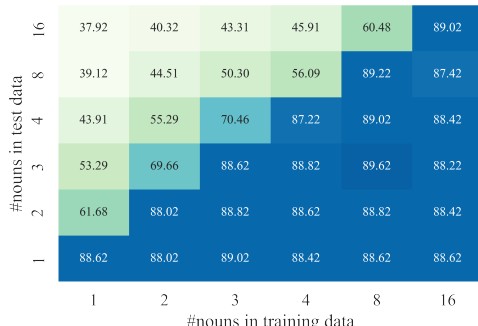

Figure 2: Syntactic accuracy of pre-trained T5 regex models (left) and baseline T5 regex models (right) trained on variations of `Regex-synthetic` with proper subsets of nouns.

behind the previous state-of-the-art. However, it is only fair to note that T5 was trained on an internet corpus, making it likely that the model has seen regular expressions during pre-training, which probably contributes to the model's high accuracy. In the next sections, we will consider FOL and LTL, where it is much more unlikely that the network has seen many instances during pre-training.

## 4.2 FIRST-ORDER LOGIC (FOL)

*Comparability to the state-of-the-art.* Singh et al. (2020) achieved an estimated semantic accuracy of $56.10\%$ on their 138K large dataset with a specialized architecture and an array of optimizations. Their dataset is similarly constructed as our 150K large dataset `FOL-mnli`, but they heuristically estimate their semantic accuracy with a matching algorithm. For best reproducibility, we thus only report on the syntactic accuracy of T5 in this paper as, at the time of writing, their dataset and code were not publically available. Their FOL formulas are represented as a reduced mapping of the `candc-boxer` output while we train on the raw output end-to-end in this work. On a held-out dataset, the fine-tuned LM achieved a syntactic accuracy of $53.91\%$, falling only 2 percentage points short of the semantically estimated state-of-the-art. On the `FOL-codesc` dataset, which was constructed to mimic code snippets, our best model achieved an accuracy of $58.98\%$ (see Figure 3 top right). It will be interesting to see how specialized approaches perform on this new dataset. Since this is a newly contributed dataset, we provide two randomly sampled successful and failed translation attempts while evaluating the best model on a held-out test set of `FOL-codesc`:

| natural language description | choose an available port |
| model prediction (correct) | fol(1,some(A,some(B,some(C,some(D,and(r1Theme(A,C), and(r1Actor(A,D),and(v1choose(A),and(n1port(C), and(a1available(B),and(r1Theme(B,C),n12thing(D))))))))))))). |
| natural language description | show start page |
| model prediction (incorrect) | fol(1,some(A,some(B,some(C,and(n1page(C),and(r1of(C,A), and(n1start(A),and(r1of(C,B),and(n1show(B),a1topic(C)))))))))))). |

*OOD-testing across datasets.* We experiment again with cross-testing in the FOL domain to report the performance of a model trained on everyday natural language (`FOL-mnli`) to the specialized domain of code (`FOL-codesc`). Note that, compared to the regex experiment, the domains considered in these datasets are much more different. A model trained on `FOL-mnli` achieved an accuracy of $31.25\%$ when tested on the code comment examples from `FOL-codesc`. Vice versa, a model achieved an accuracy of $10.55\%$. This accuracy decreases drastically for the baseline model, achieving only $19.92\%$ and $0\%$, respectively. Our experiments indicate that pre-trained language models used for code generation can translate its input into formal specifications, which formally represent their language understanding. They thus remove ambiguity and automatically formalize their input. Our long-term vision is that this additional output can be used to increase the trust in the code model's output. With the `FOL-codesc` dataset, we aim to make the first contribution toward this goal.

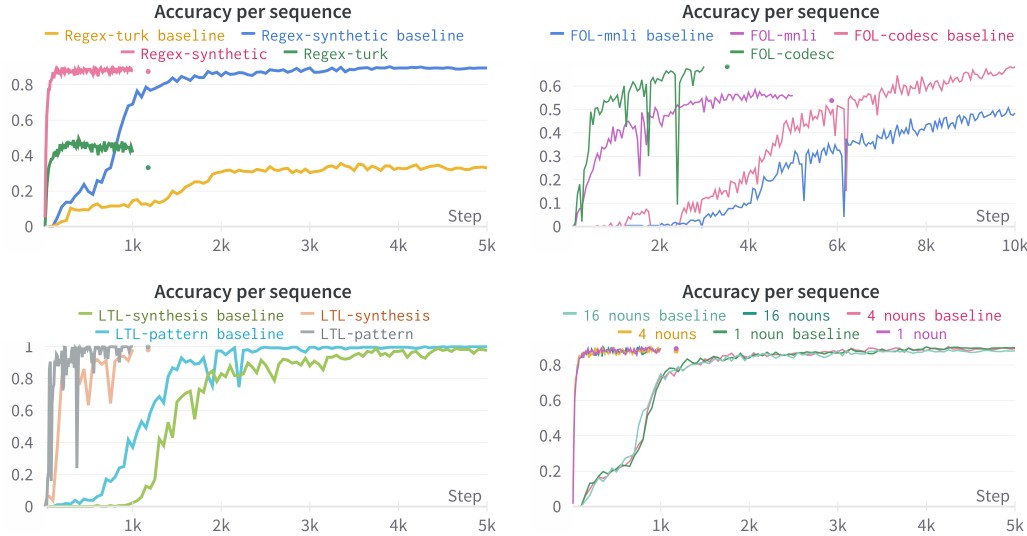

Figure 3: Respective accuracy per sequence on validation sets during training of the best performing models reported in Table 1: Regex (top left), FOL (top right), LTL (bottom left); and the accuracy per sequence for the new nouns experiment (bottom right).

## 4.3 LINEAR-TIME TEMPORAL LOGIC (LTL)

*New baseline and challenging datasets for LTL.* The language model performed well on the task of translating natural language into LTL specifications as it seems to benefit from the models generalization capabilities from pre-trained knowledge on natural language (see Figure 3 bottom left). The LTL-pattern dataset serves as a baseline, where the language model achieves an accuracy of $100.00\%$ by probably learning the underlying grammar. The LTL-synthesis dataset, however, is designed to be more challenging. It contains a combination of practical specifications to automatically synthesize hardware circuits Schmitt et al. (2021). For example:

| natural language description | Globally it is the case that if o4 holds and in the next step i0 does not hold then in the next step o4 holds and o1 does not hold until i1 does not hold or globally o1 does not hold and globally o3 holds. |
|---|---|
| model prediction (correct) | $((\Box(((o_4) \land (\bigcirc(\neg(i_0)))) \to (\bigcirc(o_4)))))$ $\land((((\neg(o_1))\,\mathcal{U}(\neg(i_1)))|(\Box(\neg(o_1)))))) \land ((\Box(o_3)))$ |

On these large instances, the language model achieves an accuracy of $87.50\%$. Failed translation attempts are, in general, due to the large size of the instances often overshooting the size limit of our language models. This experiment is especially interesting, since combining our approach with the approach of Schmitt et al. (2021) would enable the developement of a tool that synthesizes sequential hardware circuits automatically out of natural language. Final circuit predictions can then be model-checked or tested against the intermediate LTL formalization of the natural language.

*Generalization to new variable names.* We observed that the models are also able to process new variable names. Altough the models were fine-tuned on a fixed set of variables ($i_0, \dots, i_4$ and $o_0, \dots, o_4$ for LTL-synthesis, and $a, \dots, e$ for LTL-pattern) using other variables also led to correct translations. A model trained on LTL-pattern achieved an accuracy of $95.00\%$ when being tested on held-out instances where all variables were replaced with random letters from the alphabet. See Figure 1 in the introduction and the following example:

| | |
|---|---|
| natural language (ID) | Globally it is the case that if a holds then eventually a and b hold. |
| model prediction (correct) | $\Box(a \rightarrow \Diamond(a \wedge b))$ |
| natural language (OOD) | If $x$ holds infinitely often then $y$ holds infinitely often. |
| model prediction (correct) | $(\Box\Diamond x \rightarrow \Box\Diamond y)$ |

*OOD-testing across datasets.* We OOD cross-tested on the LTL datasets. Interestingly, only one of the directions showed generalization. We tested a model, trained on `LTL-pattern`, on large instances from the synthesis specifications in `LTL-synthesis`. A model trained and tested in this direction achieved an accuracy of $3.12\%$. However, in the other direction, i.e., a model trained on `LTL-synthesis` and tested on `LTL-pattern` achieved an accuracy of $37.11\%$. If we conduct the same experiment with the baseline model, the accuracy drops to $0\%$ and $7.81\%$, respectively.

*Generalization to new operator descriptions.* Lastly, we quantitatively measured the generalization of LTL models to new operator descriptions, which were kindly provided by uninvolved experts, by adding them to our translation grammar. We build two grammars, one with the additional operator descriptions and one without them (see Appendix D). Translations are then randomly chosen. A model trained on the grammar only consisting of a single translation for each operator achieved an accuracy of $53\%$ when being tested on instances generated with the enriched grammar. For example:

| | |
|---|---|
| natural language description | Always it is the case that if o2 holds then always i1 does not hold. |
| model prediction (correct) | $(\Box((o2) \rightarrow (\Box(\neg(i1)))))$ |

An additional example is the test instance hand-crafted by an expert shown in Figure 1 in Section 1, where the model recognizes "whenever" as the $\Box$-operator and the comma as the very subtle representation of an implication, both of which are not even captured by our enriched grammar. A possible use-case in this domain is the automatic formalization of software and hardware requirements from natural language to formal LTL specifications.

## 5  LIMITATIONS AND CONCLUSION

A limiting factor is that our approach still requires a GPU with enough memory to fit the language model, which detracts from its general accessibility. We set out to demonstrate the applicability of language models to a wide variety of formal domains. Nevertheless, many interesting domains are out of this work's scope but still viable targets for our approach. These include theorem proving, SQL translations, logical programming, SAT, and SMT. Another limitation is the focus on one particular class of language models. A possible further research direction is to explore the capabilities of decoder-only models such as the GPT-2 model family. Many datasets considered in this work are purely synthetic (which is only natural for the considered domains). Hence, a practical next step is encouraging experts to contribute open-source data in their respective domains. A final limitation is the unfeasibility of proper comparisons with existing works, e.g., due to unavailable datasets. With this work, we contribute to an open-source gathering of existing datasets to conduct further research.

To conclude, we conducted the first study on the generalization capabilities of fine-tuned language models to translate natural language into formal specifications, resulting in a new state-of-the-art for translating natural language into regular expressions. The benefits of fine-tuning an open-source language model are that they are easily accessible and cheap to train. We contributed two new datasets for translating natural language into First-order Logic and two new datasets for translating natural language into Linear-time Temporal Logic. We provided experiments on the generalization capabilities of the pre-trained language model T5, which serves as a baseline for further research. Our experimental results show that off-the-shelf language models can outperform approaches on existing datasets and perform well on new datasets. The language models prove themselves to be highly versatile. A unique selling point is their capability of generalizing from pre-trained knowledge of natural language, such as handling other variable names, new nouns, and new operator descriptions. We believe that the generalization capabilities of language models can be crucial in making real-world problems of translating natural language to formal specifications tractable.

## 6 REPRODUCIBILITY STATEMENT

The code, datasets, models, and notebooks for reproducing the experiments will be made publicly available once the double-blind reviewing process ends. One of the main goals of this work was to study a translation approach that is as accessible as possible. Consequently, we have used an off-the-shelf language model that is both open-source and requires significantly less memory than very large language models. Additionally, we found that, compared to training an LM from scratch, fine-tuning it has been proven robust, making this approach reproducibility-friendly.

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

# A    ABLATIONS

We reported only a selected number of trained models and conducted experiments. We tried to avoid the report of duplications, where the model shows similar behavior across all domains (such as the generalizaion to new nouns), with the exception being the OOD cross-testing, which is an interesting insight for all considered domains. The code as well as the datasets will be made publically available. We did several ablation studies while looking for the best performing models and performed a hyperparameter search for every reported model. The most influential hyperparameter for the baseline models is the learning rate. In Figure 4 we show the influence of different learning schedules on the accuracy per sequence for the `FOL-codesc` dataset. When finetuning T5 models we used a constant learning rate of $0.001$. We also experimented with larger and smaller models, since pretrained T5 models are available in different sizes. In general, the base model with 220 million parameters performed best when finetuning. Furthermore, we observed no significant increase in performance when finetuning for longer than a few thousand steps (depending on the size of the dataset between 1K to 3K steps), which takes around $1-3$ hours of training on an $A100$ for each run. Additionally, we experimented with prompting. We observed a significant (around $3\% - 5\%$) decrease in performance when omitting the prompt. Additional experiments with the prompt, for example prompting with "Translate the following *English* sentence to . . . ", lead to no significant increases or decreases in performance.

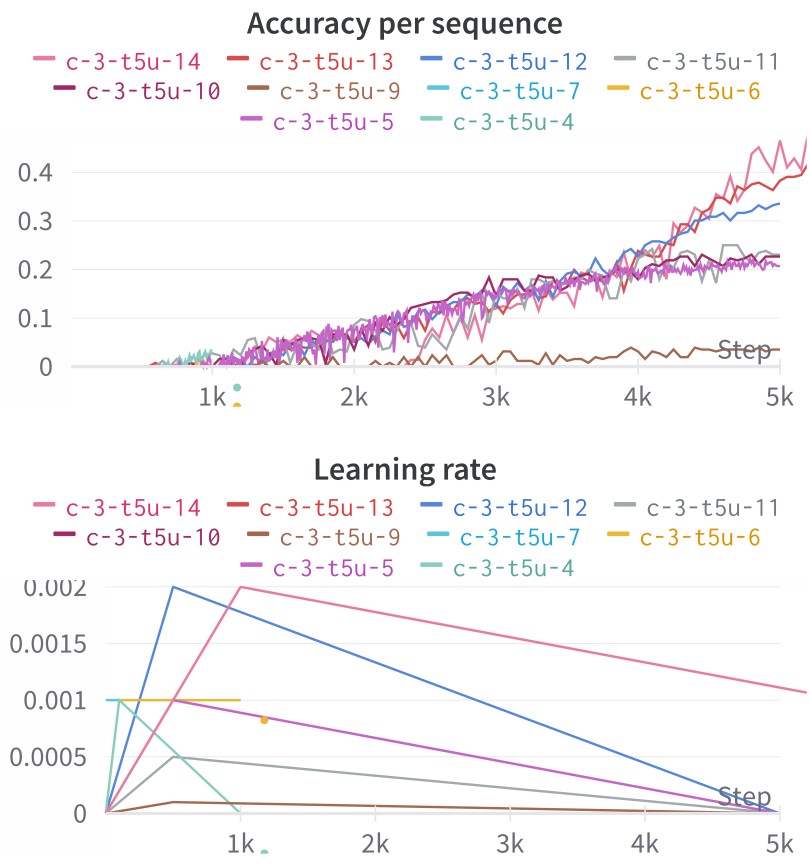

Figure 4: Sensitivity to learning rate schedule of baseline model on `FOL-codesc` dataset.

## B    DATASETS OVERVIEW

In Table 2 we give an overview of the datasets used in this work and their corresponding data source and size.

Table 2: The datasets used in this work for training and evaluation of the language models.

| Dataset | Data source | Size |
|---------|-------------|------|
| Regex-synthetic | synthesized regex Locascio et al. (2016) | 10K |
| Regex-turk | regex using amazon turk Locascio et al. (2016) | 10K |
| FOL-mnli | candc & boxer translation of mnli hypothesis | $\sim 150K$ |
| FOL-codesc | candc & boxer translation of codesc | $\sim 600K$ |
| LTL-pattern | grammar translation of specification patterns | $\sim 200K$ |
| LTL-synthesis | grammar translation of synthesis specifications | $\sim 100K$ |

## C    REGEX DEFINITION

In Figure 5 we show the regex formalism used by  Locascio et al. (2016) for creating datasets `Regex-synthetic` and `Regex-synthetic`.

| Non-Terminals | | |
|---|---|---|
| $x \mathbin{\&} y \to$ x and y | $x \mid y \to$ x or y | $\sim (x) \to$ not x |
| $.*x.*y \to$ x followed by y | $.*x.* \to$ contains x | $x\{N, \} \to$ x, N or more times |
| $x \mathbin{\&} y \mathbin{\&} z \to$ x and y and z | $x \mid y \mid z \to$ x or y or z | $x\{1, N\} \to$ x, at most N times |
| $x.* \to$ starts with x | $.*x \to$ ends with x | \b $x$ \b $\to$ words with x |
| $(x)+ \to$ x, at least once | $(x)* \to$ x, zero or more times | $x \to$ only x |

| Terminals | | |
|---|---|---|
| [AEIOU] $\to$ a vowel | $[0-9] \to$ a number | word $\to$ the string 'word' |
| [A-Z] $\to$ an uppercase letter | [a- z] $\to$ a lowercase letter | $. \to$ a character |

Figure 5: Regex syntax used in the considered datasets; taken from Locascio et al. (2016).

## D    NATURAL LANGUAGE GRAMMARS

In this section, we present the grammars that we used to construct the LTL datasets. On the highest level a formula can be, e.g., an implication, a conjunction, an equivalence or an atomic proposition. Atomic propositions as well as negated atomic propositions are represented by an `e_p`, which stands for "simple pattern". Every other subcomponent that is not an ap or a negated ap is represented by a `c_p`, which stands for "complex pattern". Binary operators like conjunction have operands that can be either easy or complex, represented by the `e_or_c` category. If the formula is complex, we need parentheses to clarify operator precedence. For instance $\square(a \wedge b)$ means that globally both a and b hold. However if we translate it directly and say "Globally a holds and b holds", we loose the meaning of the parentheses. This natural sentence could as well represent the formula $(\square a) \wedge b$. To avoid this ambiguity, we model parentheses by using the phrase "Globally it is the case that" followed by whatever the subformula is. This way it is clear that the scope of the operator extends to the entire translation of the subformula and not only to the very next part. The same principle is applied to the other unary operators such as finally and next, however not to negation as we only have negations followed by easy patterns.

The grammar with minimal domain-knowledge for a 1:1 translation beween LTL formulas and natural language is the following:

```
formula         := highest_level
highest_level   := universality | existence | implication | equivalence
                   conjunction | disjunction | until | next | e_p
universality    := "□"e_p | "□("c_p")"
existence       := "◇"e_p | "◇("c_p")"
implication     := e_or_c" → "e_or_c
equivalence     := e_or_c" ↔ "e_or_c
conjunction     := e_or_c" ∧ "e_or_c
disjunction     := e_or_c" ∨ "e_or_c
until           := e_or_c"𝒰"e_or_c
release         := e_or_c"ℛ"e_or_c
next            := "○"e_p | "○" "("c_p")"
c_p             := highest_level
e_or_c          := e_p | "("c_p")" | c_p
e_p             := ap | "!"ap
```

```
formula         := highest_level
highest_level   := universality | existence | implication |equivalence |
                   conjunction | disjunction | until | next | e_p
universality    := "Globally"e_p | "Globally it is the case that"c_p
existence       := "Eventually"e_p | "Eventually it is the case that"c_p
implication     := "if"e_or_c"then"e_or_c
equivalence     := e_or_c"if and only if"e_or_c
conjunction     := e_or_c"and"e_or_c
disjunction     := e_or_c"or"e_or_c
until           := e_or_c"until"e_or_c
release         := e_or_c"holds until"e_or_c"or forever"
next            := "in the next step"e_p | "in the next step it is the case that"c_p
c_p             := highest_level
e_or_c          := e_p | c_p
e_p             := ap "holds" | ap "does not hold"
```

In a second step, we replaced the operator descriptions with additional variations:

```
formula             := highest_level
highest_level       := universality|existence|implication|equivalence
                       conjunction|disjunction|until|next|e_p
infinitely_often    := "□(◇("e_or_c"))"
eventually_forever  := "◇(□("e_or_c"))"
universality        := "□"e_p | "□("c_p")"
existence           := "◇"e_p | "◇("c_p")"
implication         := e_or_c" → "e_or_c
equivalence         := e_or_c" ↔ "e_or_c
conjunction         := e_or_c" ∧ "e_or_c
disjunction         := e_or_c" ∨ "e_or_c
until               := e_or_c"𝒰"e_or_c
release             := e_or_c"ℛ"e_or_c
next                := "○"e_p | "○""("c_p")"
c_p                 := highest_level
e_or_c              := e_p | "("c_p")" | c_p
e_p                 := ap | "!"ap
```

```
formula             := highest_level
highest_level       := universality|existence|implication|
                       equivalence|conjunction|disjunction|until|next
                       e_p|infinitely_often|eventually_forever
infinitely_often    := "Infinitely often" e_p | "Infinitely often it is the case that" c_p
eventually_forever  := "Eventually forever" e_p |
                       "Eventually it is the case that forever" c_p
universality        := ("Globally" | "Always") e_p |
                       ("Globally it is the case that" | "Always it is the case that") c_p
existence           := ("Eventually" | "Finally") e_p |
                       ("Eventually it is the case that" | "Finally it is the case that") c_p
implication         := "if" e_or_c "then"e_or_c
equivalence         := e_or_c "if and only if" e_or_c
conjunction         := e_or_c "and" e_or_c
disjunction         := e_or_c "or" e_or_c
until               := e_or_c "until" e_or_c
release             := e_or_c "holds until" e_or_c "or forever"
next                := "in the next step" e_p | "in the next step it is the case that" c_p
c_p                 := highest_level
e_or_c              := e_p | c_p
e_p                 := ap "holds" | ap "does not hold"
```

