# OpenReview forum: "Formal Specifications from Natural Language"
_ICLR.cc/2023/Conference — Submitted to ICLR 2023_

### Official Review · Reviewer_AR6P · 2022-10-24

**Confidence:** 4
**Correctness:** 2
**Technical Novelty And Significance:** 2
**Empirical Novelty And Significance:** 2
**Recommendation:** 5

**Clarity, Quality, Novelty And Reproducibility:**

The presentation is clear and easy to follow. The authors promised the release of code and dataset for reproducibility.

For novelty, see comments above.

**Strength And Weaknesses:**

Strength

+ The empirical evaluation validates the approach showing that the performance compares favorably to that of previous approaches, though limited to the problem of tranlation from natural language to regular expressions.

+  The study of out-of-distribution performance by having two categories of datasets for each problem is nice to have, which helps understand the capability of LLMs on logic tasks.

Weakness

- While I appreciate the claimed SOTA (of 2% lead) on translation to regex, the overall contribution of the paper seems a bit pale. The paper claimed two new datasets being part of the contribution, but did not provide proper experiments justifying the fairness of them.

- The analysis of generalizability is limited. It is not that surprising that language models had no problems recognizing new nouns and variable names, as well as equivalent descriptions of operators, since they have seen enough on the internet (which could also contain some problems in the experiments.) What would be interesting to see, for example, could be the performance with respect to the complexity of a formula (e.g., the number of logical connectives).

- It is unclear how the accuracies used for FOL and LTL are defined. It is understandable that semantic equivalence for FOL is difficult, but how is syntactic accuracy done in the paper? Do you simply count an exact match as correct?

- More baselines could have been added to support the claim of the paper. For example, have you tried few-shot learning with T5? If few-shot learning (with reasonably good example prompts) works, then it probably means that the language model has already seen enough translation examples during training.

**Summary Of The Paper:**

The paper investigates the application of language models to translation from statements in natural language to formal languages including regular expressions, first order logic and linear temporal logic. The authors fine-tuned a pre-trained language model (T5) on three groups of datasets, each corresponding to the problem of translation from natural language to one formal language, and reported that the fine-tuned model outperforms the SOTA of translation to regular expressions. The authors also observed that the model could generalize to out-of-distribution instances to some extent, as well as instances where changes are made to variable names, nouns and operator descriptions.


**Summary Of The Review:**

The authors presented an interesting observation that language models are capable of translating natural languages to formal languages when fine-tuned on target problems. However, I think the other contributions claimed in the paper need to be supported by more content before the paper can be accepted.

---

### Official Review · Reviewer_NwtW · 2022-10-25

**Confidence:** 3
**Correctness:** 3
**Technical Novelty And Significance:** 1
**Empirical Novelty And Significance:** 2
**Recommendation:** 3

**Clarity, Quality, Novelty And Reproducibility:**

Clarity: 3/5
Pros:
1. The definition of the formal languages are included which makes the output space very clear
2. The included examples of each task demonstrate the difficulty of the datasets well
Cons:
1. The dataset generation process is not well elaborated.

Quality: 2/5
Pros:
1. The performed evaluations are quite comprehensive.
Cons:
1. A qualitative comparison is not included among different datasets to illustrate your advantage.
2. The contribution of the dataset generation process is missing.
3. Unfair comparison. The model size for this work seems to be much larger than the previous Regex conversion SOTA. However, a model size comparison is not included in the comparison.

Originality: 1/5
This work heavily relies on the previous ones and does not include enough innovative ideas.


**Strength And Weaknesses:**

Strengths:
1. The study is performed on multiple formal languages, including Regex, FOL, and LTL. This demonstrates that the model can adapt to different formal languages with a similar pipeline.
2. The examples for Regex and FOL show the mapping is non-trivial.

Weakness:
Two aspects can be taken to view the weakness.
Aspect 1: Consider this paper as a dataset paper. Both FOL-mnli and FOL-codesc are generated using previous techniques; the LTL-pattern and LTL-synthesis are also not well addressed for their difference from the literature. Further, the natural language part of LTL-synthesis is quite synthetic and template based.
Aspect 2: Consider this paper as a methodology paper. The methodology is not that deep and does not contain deep technical contributions. One word that the authors compare to is folio, which the authors claim they adopt a similar methodology as theirs. However, the folio work is indeed a dataset paper, and all the few-shot GPT tasks are simply baselines for demonstration purposes.


**Summary Of The Paper:**

The authors propose a generalizable architecture using large language models as a backbone to translate the natural language into formal specifications.
Contributions:
1. First off-the-shelf fine-tuned language model for natural language to formal specification.
2. New datasets for translating natural language into FOL and LTL
3. Generalizability capacity

**Summary Of The Review:**

The authors propose a generalizable architecture using large language models as a backbone to translate the natural language into formal specifications. However, this work heavily relies on the previous literature and does not include enough innovative ideas.

---

### Official Review · Reviewer_i1Lj · 2022-10-25

**Confidence:** 4
**Correctness:** 3
**Technical Novelty And Significance:** 1
**Empirical Novelty And Significance:** 2
**Recommendation:** 3

**Clarity, Quality, Novelty And Reproducibility:**

I believe the paper is hard (or even impossible) to follow for readers that are not already familiar with the syntax and semantics of the considered formal languages.

While some technical details are missing, it is likely that the work could be reproduced.

Novelty is restricted to the construction of new datasets that seem to be largely unrepresentative of practicall interesting tasks.

Notes:
* page 1: "have relied on training (often over-engineered) neural networks": please do not pass judgement on related work without providing any evidence.
* page 1: "The natural questions arise": it is unclear why these are natural questions. Please rephrase.
* page 2: "(a|b)* reads as 'all sequences with no symbols ...'": this is not "reads" but a full interpretation. Please rephrase.
* Sect. 2: the first two paragraphs seem to have been swapped without appropriate editing (the first one starts with "Similarly to FOL", which is discussed below)
* Sect. 3.1: I believe the "2 or more times" example suffers from misquoting, and {} vanished?
* Sect. 3.2: Please provide some examples of the generated FOL formulae here. The examples shown in Sect. 4.2 are not understandable.
* Sect. 3.3: I don't think this section makes sense to anyone who's not already familiar with LTL. In particular, references to "timesteps" or "cycles" make no sense without explaining the underlying notion of a (potentially finite) state system.
* Sect. 4: "We fine-tuned [...] for arond 1 hour": this is not helpful. How did you decide to stop? Wallclock time? Number of training steps? Validation results?
* Sect 4.3: Given that the "natural language" has been automatically generated from the formula, wouldn't it be possible to write a deterministic parser for this, following the grammar on p19? Why do the models not achieve 100% accuracy on this dataset?


**Strength And Weaknesses:**

* (+) Evaluating the ability to translate natural language into formal languages is important for better human/computer interfaces.
* (-) The generated datasets are unrepresentative of _natural_ language. (see the examples in Sect 4.2 and 4.3)


**Summary Of The Paper:**

The ability of language models to translate from natural language into formal languages is studied. Concretely, regular expressions, first-order logic and linear-time temporal logic are considered as target languages. Four new synthetic datasets are created to measure success on the latter two languages.
Experiments indicate that off-the-shelf fine-tuned Transformer models can handle the chosen tasks reasonably well. Some efforts are made to consider some axes of generalization in the experiments.


**Summary Of The Review:**

It is unclear what insights the reader should take away here, and the authors do not outline future work. Overall, the contribution is limited to running experiments on four new datasets that are all problematic in their own ways, and are not connected to pre-existing tasks. Hence, I do not think this paper is of substantial interest to the ICLR community and would hence recommend rejection. It may be better suited for a workshop more focused on neural models for formal languages.

---

### Official Review · Reviewer_xRr1 · 2022-10-31

**Confidence:** 4
**Correctness:** 3
**Technical Novelty And Significance:** 2
**Empirical Novelty And Significance:** 3
**Recommendation:** 5

**Clarity, Quality, Novelty And Reproducibility:**

Overall, the paper and method are clearly written. The paper is of limited technical novelty, however, the authors do investigate an interesting problem empirically. The results are reproducible if the datasets for FOL and LTL are released.

**Strength And Weaknesses:**

The paper is well written and well motivated, the authors clearly describe the problem and the approach for data generation and evaluation. However, the experimental section is a bit limited and the authors seem to draw strong conclusions from experiments on small datasets (see questions below).
- The small version of T5 model is used for the baseline experiments, but the base version is used for fine-tuning experiments. Is the comparison fair? How do the baseline experiments compare to fine-tuning the small version?
- The generalization test for regex is investigated on a very small set, what can really be inferred from this experiments? Similarly for investigating the generalization to new operator descriptions for LTL, the set of alternatives is very small.
- For generalization to new operator descriptions for LTL, the performance of the model goes from 100% to 53%. Where does it fail?

**Summary Of The Paper:**

The paper focuses on the problem of translating natural language into formal specifications, in particular, regular expressions (regex), first order logic (FOL), and linear temporal logical (LTL). Instead of training deep models from scratch for this problem, the authors investigate whether fine-tuning pretrained language models achieves similar performance, and how well it generalizes to for example new variable names or operator descriptions. The authors consider six datasets in their work, two for each domain: they use existing benchmark datasets Regex-synthetic and Regex-turk, and synthetically generate the datasets for FOL (FOL-mnli and FOL-codesc) and LTL (LTL-pattern and LTL-synthesis) which are also contributions of the paper.

The authors use the T5 model and compare training from scratch to fine-tuning pretrained T5 model for their experiments. On the regex datasets, they report a 6% improvement over SOTA results on both synthetic and turk datasets when fine-tuning the T5 model. On FOL-mnli, they also show a 7% improvement when fine-tuning compared to training from scratch. On LTL datasets, training from scratch and fine-tuning gives similar results. They evaluate generalization by considering nouns that were not present during fine-tuning for regex, new variable names and operator descriptions for LTL. They also conduct cross testing, where they train on one dataset for a domain and evaluate on the other dataset. They show that while the models have acceptable performance for regex, their performance drastically decreases for FOL and LTL datasets.

**Summary Of The Review:**

The problem investigated is interesting. However, given the questions raised above, and parts of the evaluation being limited, I don't believe the paper is ready as is.

---

### Author Response · Authors · 2022-11-18
**General Rebuttal**

We genuinely thank the reviewers for taking the time to review our submission and for their valuable suggestions on improving the paper's presentation.
There are a few points, however, where we disagree with the assessment of our submission:

1) The datasets are unrepresentative of natural language:

The datasets for regex and FOL are based on natural language snippets and standard datasets.
Considering the synthetically generated LTL datasets: The contribution of this paper is the observation that after fine-tuning, a) the LM learns the complex semantics of the specification languages from the synthetic data and b) transfers their “knowledge” of natural language from pre-training. This was far from obvious when starting this project, as either: “forgetting” during fine-tuning or poor accuracy (or both) of the off-the-shelf LM would be likely.

2) Methodology:

The low technical entry barrier of our approach is a significant benefit over previous work. It is a specification language agnostic, open-source, simple, and resource-saving approach. As mentioned above, it is far from obvious that the off-the-shelf LM would be competitive due to either “forgetting” knowledge of the natural language during fine-tuning or resulting in poor accuracy compared to previous approaches engineered to particular specification languages.

3) Not interesting for the community and not practical:

We disagree. We consider the findings of this paper valuable for the formal methods and ml communities. All specification languages considered in this paper find application in practice. The observations in this paper open novel research directions in the intersection of formal approaches in computer science and deep learning on natural language.

---

### Decision · Program_Chairs · 2023-01-20

**Decision:**

Reject

**Justification For Why Not Higher Score:**

There was no attempt at a rebuttal.

**Justification For Why Not Lower Score:**

This is the lowest score.

**Metareview: Summary, Strengths And Weaknesses:**

This paper received scores of 3,3,5,5 and no rebuttal was given.